# Lactate to Albumin Ratio and Mortality in Patients with Severe Coronavirus Disease-2019 Admitted to an Intensive Care Unit

**DOI:** 10.3390/jcm13237106

**Published:** 2024-11-24

**Authors:** Stelios Kokkoris, Aikaterini Gkoufa, Dimitrios E. Katsaros, Stavros Karageorgiou, Fotios Kavallieratos, Dimitrios Tsilivarakis, Georgia Dimopoulou, Evangelia Theodorou, Eleftheria Mizi, Anastasia Kotanidou, Ioanna Dimopoulou, Christina Routsi

**Affiliations:** First Department of Critical Care Medicine and Pulmonary Services, National and Kapodistrian University of Athens Medical School, Evangelismos Hospital, 45-47 Ipsilantou Street, 10676 Athens, Greece; skokkoris2003@yahoo.gr (S.K.); katergouf@yahoo.gr (A.G.); de.katsaros@gmail.com (D.E.K.); stavros99k@gmail.com (S.K.); kavallieratos13@gmail.com (F.K.); tsilivarakisd@gmail.com (D.T.); ginadim@outlook.com (G.D.); evaggeliatheodorou@gmail.com (E.T.); eleftheria.mizi@yahoo.com (E.M.); akotanid@gmail.com (A.K.); idimo@otenet.gr (I.D.)

**Keywords:** lactate, albumin, COVID-19, SARS-CoV-2, sepsis, outcome, intensive care unit

## Abstract

**Aim:** This study sought to evaluate the effectiveness of lactate/albumin ratio for ICU mortality prediction in a large cohort of patients with severe Coronavirus Disease-2019 (COVID-19) admitted to an intensive care unit (ICU). **Methods:** This is a single-center retrospective cohort study of prospectively collected data derived from the COVID-19 dataset for all critically ill patients admitted to an academic ICU. Data were used to determine the relation between lactate/albumin ratio and other laboratory parameters measured on the first day of the ICU stay and to evaluate the prognostic performance for ICU mortality prediction. **Results:** A total of 805 ICU patients were included, and the median age (IQR) was 67 (57–76) years, with 68% being male. ICU mortality was 48%, and the median lactate/albumin ratio was 0.53 (0.39–0.59). A survival analysis showed that patients with higher lactate/albumin ratio values had significantly lower survival rates (Log Rank *p* < 0.001). A multivariable analysis revealed that the lactate/albumin ratio was an independent risk factor for ICU mortality with a hazard ratio of 1.39 (CI: 1.27–1.52). The lactate/albumin ratio showed a receiver operating characteristics area under the curve (ROC-AUC) value to predict ICU mortality significantly higher than that of lactate alone (0.71 vs. 0.68, DeLong test *p* < 0.001). The optimal lactate/albumin ratio cut-off for predicting ICU mortality was 0.57, with 63% sensitivity and 73% specificity. A subgroup analysis revealed that the lactate/albumin ratio was significantly associated with mortality across different patient groups, including age and sex categories, and those with or without hypertension and coronary heart disease. **Conclusions**: Lactate/albumin ratio is a reliable prognostic marker in critically ill COVID-19 patients and could predict ICU mortality more accurately than lactate alone.

## 1. Introduction

In critically ill patients, inadequate oxygen supply to tissues results in anaerobic glycolysis and consequently increased lactate production [1,2]. Elevations in blood lactate concentrations, i.e., hyperlactatemia with or without acidosis, is considered a sign of organ hypoperfusion and tissue hypoxia, signifies acute circulatory failure, and is closely related to the severity of the underlying condition and clinical outcome [1,2,3]. However, hyperlactatemia does not always reflect tissue hypoxia. Apart from anaerobic production, lactate release can result from other non-hypoxic factors, such as increased aerobic glycolysis secondary to activation of the stress response [4,5]. Especially in patients with sepsis, hyperlactatemia can also be associated with an alteration in oxygen extraction that has been attributed to microcirculatory dysfunction and to impaired cellular oxygen utilization, as well as to changes in lactate clearance [6,7]. Although the source of lactate in sepsis is a matter of debate, increased blood lactate levels are currently used worldwide for early diagnosis, management, and risk stratification of patients with sepsis and/or septic shock [8].

In addition to lactate, albumin, which is known to be a negative acute phase protein produced by the liver, in the setting of acute illness is downregulated, and its concentration decreases. As a result, hypoalbuminemia upon hospital admission is common in critically ill patients. Among other factors, such as malnutrition, nitrogen balance, or renal replacement therapy, a decreased serum albumin level is mainly attributable to systemic inflammation, resulting in increased vascular permeability, leading thus to a greater capillary leakage of albumin in the interstitial space [9,10]. Accordingly, hypoalbuminemia is considered to be a reliable indicator of the severity of the inflammation state, and it has been associated with poor outcomes [11].

Previous research has shown that the ratio between lactate and albumin could provide further information as an early prognostic marker for critically ill sepsis patients [12,13,14,15,16,17], as well as in patients with sepsis in the emergency department [18,19], having better prognostic performance than lactate alone in most of the studies. Coronavirus Disease-2019 (COVID-19) caused by the severe acute respiratory syndrome Coronavirus 2 (SARS-CoV-2), as a cause of viral sepsis [20], could justify the possibility of similar responses to lactate and albumin values. Indeed, severe COVID-19 has been associated with immunological and vascular changes [21], as well as alterations in many acute phase proteins including albumin [22]. Rapid albumin loss has been described in critically ill patients with COVID-19 during the early phase of ICU admission, regardless of outcome [22,23].

Regarding the lactate level, in the context of severe COVID-19, the majority of patients present normal lactate values on admission to the intensive care unit (ICU). However, a significant percentage of patients have elevated levels; these patients exhibit a significantly higher ICU mortality risk [24,25,26].

The ratio between lactate and albumin has not been studied in critically ill patients with COVID-19. Therefore, we evaluated the prognostic performance of the lactate to albumin ratio on ICU mortality in a large cohort of critically ill patients admitted to our ICU during the pandemic. We hypothesized that the lactate/albumin ratio is a better prognostic marker than lactate alone in the prediction of mortality among critically ill patients admitted to the ICU due to COVID-19.

## 2. Methods

### 2.1. Study Design

This is a single-center retrospective cohort study of prospectively collected data derived from the COVID-19 dataset (formed in March 2020) [27] for all critically ill patients admitted to the university ICU at “Evangelismos” Hospital, a tertiary care center in Athens, Greece, between March 2020 and April 2022. All patients suffered from severe SARS-CoV-2 infection, confirmed by a real-time reverse transcriptase-polymerase chain reaction assay of nasopharyngeal swab specimens. Patients who died within 24 h post ICU admission, as well as patients who received an infusion of albumin solution before or on admission to the ICU, were excluded.

### 2.2. Data Collection

The collected data encompassed demographics, laboratory tests, illness severity upon ICU admission, comorbidities, need for mechanical ventilation, vasoactive agents use, vaccination status, remdesivir and/or dexamethasone treatment, requirement for continuous renal replacement therapy (CRRT), length of ICU stay, and ICU clinical outcome. Routine laboratory tests included red blood cell (RBC) and white blood cell (WBC) counts, neutrophil and lymphocyte counts, hemoglobin, hematocrit, platelet count, C-reactive protein (CRP), lactate dehydrogenase (LDH), creatinine, highly sensitive cardiac troponin I (Hs-cTnI), aspartate aminotransferase (AST), alanine aminotransferase (ALT), bilirubin, sodium, glucose, d-dimers, fibrinogen, lactate, albumin, and PaO_2_/FiO_2_ ratio.

The illness severity was evaluated using the Acute Physiology and Chronic Health Evaluation (APACHE) II [28] and the Sequential Organ Failure Assessment (SOFA) [29] scoring systems, calculated on the first day of ICU admission. Comorbidities included hypertension, diabetes mellitus, obesity, cardiovascular disease, chronic pulmonary disease, chronic kidney disease, and active malignancy. To account for comorbidities, the Charlson Comorbidity Index was calculated [30]. Shock was defined as hypotension (systolic blood pressure < 90 mm Hg and/or mean arterial pressure < 65 mm Hg), persisting despite adequate volume resuscitation, requiring the administration of vasoactive agents [31].

The lactate /albumin ratio was calculated by dividing the arterial lactate level by the serum albumin value. In addition, the neutrophil to lymphocyte ratio (NLR) was calculated by dividing the absolute neutrophil count by the absolute lymphocyte count. Variables not routinely measured, such as ferritin or cytokines, were not included in the analysis. All variables mentioned above were collected within 24 h of patients’ ICU admission. If the patient had multiple measurements within this time period, data from the initial measurement were used. The collection of anonymized data for the study was approved by the Hospital Ethics Committee (Protocol Number 116/2021).

### 2.3. Statistical Analysis

Continuous variables were expressed as median and interquartile range (IQR), while categorical variables were presented as proportions. Continuous variables were analyzed using the Mann–Whitney U test or Kruskal–Wallis test. To handle missing data, we employed multiple imputations, implemented through the ‘mice’ package v. 3.16.0 in R software. All variables used in the analyses had less than 5% of missing values, except for d-dimers with 19%.

Correlations between variables were estimated using Spearman’s Rho coefficient. Cox proportional hazards models were used to calculate the hazard ratio (HR) and the 95% confidence interval (CI) of the lactate/albumin ratio for ICU all-cause mortality, with adjustments made for multiple variables. The multivariate analysis included the following variables: age, sex, NLR, Hb, PLT count, sodium, creatinine, AST, ALT, LDH, hs-cTnI, CRP, fibrinogen, d-dimers, PaO_2_/FiO_2_ ratio, presence of shock, vaccination status, CRRT, remdesivir, dexamethasone, hypertension, diabetes mellitus, obesity, cardiovascular disease, chronic pulmonary disease, chronic kidney disease, and active malignancy. The proportional hazards assumption was checked using the Schoenfeld residuals. Kaplan–Meier survival analysis was used to estimate the ICU mortality based on the lactate/albumin ratio quartiles. Additionally, we analyzed the nonlinear association between the lactate/albumin ratio and ICU all-cause mortality using a restricted cubic spline regression model with four knots. The Hosmer–Lemeshow goodness-of-fit test was used to evaluate whether the logistic regression model was a good fit for the data. Receiver operating characteristic (ROC) curves were used to determine the cutoff value of the lactate/albumin ratio and to evaluate the predictive power of the lactate/albumin ratio, lactate, albumin, and APACHE II score for ICU all-cause mortality. An optimal cut-off was calculated by means of Youden’s index. Improvement in APACHE II score’s (reference model) predictive ability for ICU outcome after the addition of the lactate/albumin ratio was evaluated using the net reclassification improvement (NRI) method and AUC’s difference (ΔAUC). Further stratified analyses were conducted based on age (<65 and ≥65 years), sex, hypertension, and coronary heart disease to assess the consistency of the prognostic value of the lactate/albumin ratio for outcome. Interactions between the lactate/albumin ratio and stratification variables were examined using likelihood ratio tests. All analyses were performed using R software (version 4.3.3), and a two-tailed *p*-value < 0.05 was considered statistically significant. We used the following R statistical packages: dplyr v. 1.1.4, ggplot2 v. 3.5.1, mice v. 3.16.0, gtsummary v. 2.0.2, corrplot v. 0.94, survival v. 3.6-4, survminer v. 0.4.9, pROC v. 1.18.5, splines v. 4.4.1, ggrcs v. 0.4.1, caret v. 6.0-94, nricens v. 1.6, and gridEXTRA v. 2.3.

## 3. Results

### 3.1. Baseline Characteristics

A total of 805 patients were included in the final analysis. Of them, 548 (68%) were males, and their median (IQR) age was 67 (57–76) years. APACHE II score was 14 (11–19) and SOFA score was 7 (4–9). The most frequent comorbidities were hypertension (41%), diabetes (25%), and cardiovascular disease (25%). All-cause ICU mortality in the entire population was 48%, and CRRT was required by 30% of patients; 61 (7.6%) patients were fully vaccinated, 637 (79%) received dexamethasone, and 391 (49%) received remdesivir. Median lactate/albumin ratio was 0.53 (0.39–0.79), Table 1. Lactate/albumin ratio, lactate, albumin, hs-cTnI, d-dimers, fibrinogen, age, APACHE II and SOFA scores, Charlson comorbidity index, as well as the presence of cardiovascular disease, were significantly higher in survivors as compared to non-survivors (Table 1). Baseline characteristics of the cohort according to the lactate/albumin ratio quartiles are shown in Table 2. ICU mortality, CRRT need, mechanical ventilation on admission, age, APACHE II score, SOFA score, and Charlson comorbidity index, as well as the presence of cardiovascular disease, were significantly different across the lactate/albumin ratio quartiles. Figure 1 depicts the boxplots of the lactate/albumin ratio according to age quartiles. The lactate/albumin ratio was significantly increased in the fourth quartile compared with the lower age quartiles (4 vs. 1, *p* < 0.001; 4 vs. 2, *p* = 0.007).

### 3.2. Correlations of Lactate/Albumin Ratio with Other Variables

Figure 2 demonstrates the correlation matrix of the lactate/albumin ratio with other variables. The lactate/albumin ratio was significantly correlated with age (r = 0.14, *p* < 0.001), Charlson comorbidity index (r = 0.20, *p* < 0.001), APACHE II (r = 0.43, *p* < 0.001), and SOFA (r = 0.40, *p* < 0.001) scores, NLR (r = 0.09, *p* = 0.002), PLT count (r = −0.13, *p* < 0.001), creatinine (r = 0.11, *p* = 0.001), LDH (r = 0.42, *p* < 0.001), Hs-cTnI (r = 0.30, *p* < 0.001), CRP (r = 0.07, *p* < 0.02), fibrinogen (r = −0.19, *p* < 0.001), and d-dimers (r = 0.12, *p* < 0.001).

### 3.3. Survival Analysis

Figure 3 illustrates Kaplan–Meier survival curves for the lactate/albumin ratio quartiles. A log-rank test revealed significantly lower survival probability for the higher lactate/albumin ratio quartiles (*p* < 0.001). In a univariable Cox proportional hazards analysis, the lactate/albumin ratio was significantly associated with ICU mortality (HR = 1.6, CI: 1.4–1.7). Furthermore, a multivariable model adjusted for multiple parameters revealed the lactate/albumin ratio to be an independent risk factor for ICU mortality (HR = 1.39, CI: 1.27–1.52), Table 3. Figure 4 illustrates a restricted cubic spline regression model, revealing a nonlinear relationship between the lactate/albumin ratio level and the ICU all-cause mortality in the aforementioned multivariable model (*p* for nonlinear = 0.025).

### 3.4. ROC Curves Analysis

Figure 5 depicts the ROC curves of the APACHE II score, lactate/albumin ratio, lactate and albumin for ICU mortality prediction, with AUCs of 0.83 (95% CI: 0.80–0.86), 0.71 (0.68–0.75), 0.68 (0.64–0.72), and 0.69 (0.65–0.73), respectively. The DeLong test demonstrated that the AUC of lactate/albumin ratio was significantly higher than that of lactate (*p* < 0.001). An improvement in the APACHE II score’s (reference model) predictive ability for ICU outcome after the addition of the lactate/albumin ratio was evaluated using NRI and ΔAUC (ΔAUC = 0.0049, DeLong’s *p* = 0.0536, and NRI = 0.0063, CI: −0.1232–0.040). The non-significant ΔAUC, as well as the NRI’s CI crossing zero, indicate that there is no strong evidence to support the notion that the addition of the lactate/albumin ratio to the APACHE II score is better or worse in terms of reclassification improvement.

The predictive characteristics of the admission lactate/albumin ratio for ICU outcome at three representative threshold values are shown in Table 4. An optimal cut-off value of the lactate/albumin ratio of 0.57 had a sensitivity of 63% and a specificity of 73% for ICU mortality prediction.

### 3.5. Subgroup Analyses

We also performed a risk stratification analysis of the lactate/albumin ratio for the primary endpoint across multiple subgroups, such as age, sex, hypertension, and cardiovascular disease. The lactate/albumin ratio was significantly associated with increased ICU all-cause mortality in subgroups defined by age < 65 years, age ≥ 65 years, female and male sex, presence and absence of hypertension, as well as presence and absence of cardiovascular disease. Significant interactions were found between the lactate/albumin ratio and sex, as well as between the lactate/albumin ratio and cardiovascular disease in the subgroup analyses (all *p* for interaction < 0.05). Figure 6 shows the forest plot of the subgroup analyses for the association of the lactate/albumin ratio with ICU mortality.

## 4. Discussion

In this study, we evaluated the effectiveness of the lactate/albumin ratio for mortality prediction in a large cohort of patients with severe COVID-19 admitted to the ICU. The main findings are the following: (i) patients with high lactate/albumin ratio on the first day of ICU admission had significantly lower survival rates, (ii) the lactate/albumin ratio was an independent risk factor for ICU mortality, (iii) ROC curve analysis showed that the lactate/albumin ratio was superior in outcome prediction than lactate alone. These results underscore the utility of the lactate/albumin ratio as a predictive marker for ICU outcome in COVID-19 critically ill patients.

The prognostic significance of the lactate/albumin ratio in septic patients has been investigated in several studies [12,13,14,15,16,17,18,19]. Wang B. and colleagues [12] were the first to demonstrate the lactate/albumin ratio as an independent predictor of mortality in patients with severe sepsis and septic shock. Since then, the majority of the subsequent studies confirmed the better prognostic performance of the lactate/albumin ratio compared to a single lactate measurement in predicting ICU mortality [13,14,15,16,17,18], although elsewhere the performance of the lactate/albumin ratio was equivalent to lactate alone in predicting mortality in a large cohort of ICU patients [19].

However, even though the aforementioned recent studies have focused on the use of the lactate/albumin ratio to predict mortality in critically ill patients with sepsis, this index has not been studied in patients with COVID-19, which represents a type of viral sepsis [20]. To the best of our knowledge, the present study is the first to examine the prognostic performance of the lactate/albumin ratio in patients admitted to the ICU due to COVID-19. We showed that at admission to the ICU, both lactate and albumin as well as their ratio exhibited an acceptable performance; however, the lactate/albumin ratio appeared to be superior for predicting mortality in these patients, as reflected by a higher AUC compared to that of lactate alone. These findings are in accordance with those of the previous studies in non-COVID populations, mentioned above [12,13,14,15,16,17,18].

To interpret the higher performance of the lactate/albumin ratio compared to each parameter alone, one could take into account that both parameters represent sepsis severity and are useful prognostic markers for clinical outcome in critically ill patients [8,11]. Therefore, the better ability of their combination in predicting ICU mortality in patients with severe COVID-19 seems to be expected and reasonable.

In addition, our study examined the utility of the lactate/albumin ratio in various subgroups inside the cohort of the COVID-19 critically ill patients, defined by age, sex, and history of hypertension and cardiovascular disease. These analyses revealed that the prognostic value of the lactate/albumin ratio was consistent across these subgroups, after adjusting for all covariates, with notable interactions between the lactate/albumin ratio and sex, as well as history of cardiovascular disease.

The role of the lactate/albumin ratio as a prognostic marker is further supported by comparing its performance with other biomarkers. Jeong et al. [32] compared the lactate/albumin ratio with the red cell distribution width/albumin ratio and found the lactate/albumin ratio to be equally effective in predicting 28-day mortality in critically ill patients with pneumonia requiring mechanical ventilation. On the contrary, Ren et al. [33] have reported a nonlinear relationship between the lactate/albumin ratio and in-hospital mortality in ICU patients with acute respiratory failure of various etiologies. By using a 2-segment linear regression model, the inflection point was calculated to be 4.46. On the left side of the inflection point, LAR was positively associated with in-hospital mortality, whereas an inverse association between LAR and in-hospital mortality was observed on the right side of the inflection point.

In a very recent work by Wang HX and colleagues [34], the lactate/ albumin ratio was an independent predictive factor for 28-day overall mortality in patients with acute respiratory distress syndrome (ARDS). Interestingly, the ROC-AUC of the lactate/albumin ratio was equal to that of our study (0.71) and provided significantly higher discrimination compared with lactate or albumin alone, similar to our findings. Since almost all patients of our cohort presented COVID-19-related ARDS, the consistency of the results of both studies is not surprising.

Furthermore, it should be noted that the lactate/albumin ratio has also shown promise in specific subgroups of critically ill patients, such as those with heart failure [35], acute myocardial infarction [36], acute kidney injury undergoing renal replacement therapy [37], burn patients [38], as well as in pediatric septic shock patients with underlying chronic disease [39], suggesting that the lactate/albumin ratio is not only a general marker of critical illness severity, but also has condition-specific utility.

As shown in Figure 5, in the present study, the APACHE II score yielded a better AUC compared to the lactate/albumin ratio, whereas in other relevant articles, the AUC values of the severity scores that were used either are not reported [13,14,15,17], or they present a similar performance with the lactate/albumin ratio [34,40] There is only one study by Wang et al. [12] showing the lactate/albumin ratio to be superior to the APACHE II score in predicting mortality, although these findings are questionable due to the small size of the patient population compared to that of the remaining articles. This better performance is not unexpected for the APACHE II score, since it is a well-established and the most commonly used scoring system over several decades [28]. However, it should be noted that, as it is based on the quantitative classification of twelve variables, it is time consuming, requiring preprinted charts or online calculators. On the contrary, the lactate/albumin ratio is a biomarker that is very easy to obtain, as it is already validated in large cohorts of unselected critically ill patients as an early prognostic marker of ICU mortality.

Certain limitations include the retrospective nature of this single-center study, factors that may limit the ability to draw definitive conclusions about causality as well as the lack of information on the trajectory of the lactate/albumin ratio and its association with outcomes. Thus, the impact of dynamic changes in the ratio on prognosis has not been assessed. In addition, albumin levels can be influenced by factors beyond critical illness, such as nutritional status, liver function, and fluid balance. This variability can introduce confounding factors into the lactate/albumin ratio measurements and affect its reliability as a prognostic tool. Particularly, the inverse correlation between albumin and age, possibly indicating malnutrition, observed in the present study, along with the increased mortality in older adults as shown in this study and elsewhere [27,41], suggest an obscure contribution of low albumin level to the lactate/albumin ratio. However, the large size of our sample, including all patients admitted to our ICU due to COVID-19 over the pandemic, minimizes potential sources of confusion.

In conclusion, the lactate/albumin ratio at ICU admission, easily obtained from routine laboratory tests, performs better than lactate alone, and it could serve as a reliable indicator for the prognosis of critically ill COVID-19 patients. Further research should focus on refining lactate/albumin ratio risk models and exploring its role in guiding therapeutic interventions. The integration of the lactate/albumin ratio into established clinical scoring systems may further enhance its utility in predicting outcomes and improving patient care.

## Figures and Tables

**Figure 1 jcm-13-07106-f001:**
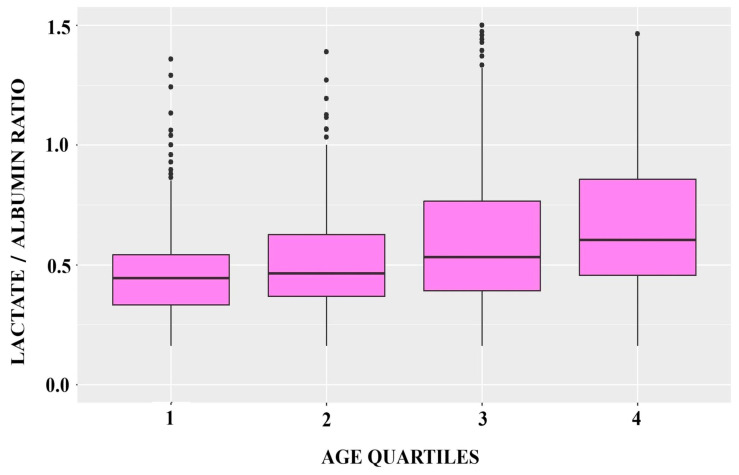
Boxplots of lactate/albumin ratio according to age quartiles.

**Figure 2 jcm-13-07106-f002:**
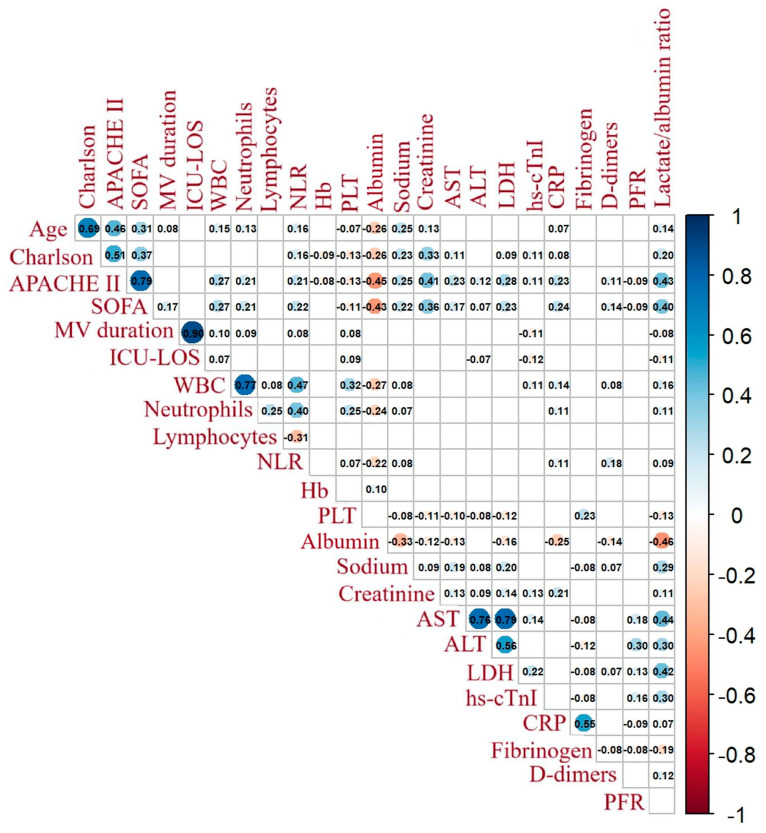
Correlation matrix of lactate/albumin ratio with other variables. Numbers inside squares represent Spearman’s rho correlation coefficients. Blank squares denote non-significant correlations, e.g., *p* > 0.05. Abbreviations: APACHE, acute physiology and chronic health evaluation; SOFA, sequential organ failure assessment; MV, mechanical ventilation; ICU–LOS, intensive care unit–length of stay; WBC, white blood cell; NLR, neutrophil to lymphocyte ratio; Hb, hemoglobin; PLT, platelets; AST, aspartate aminotransferase; ALT, alanine aminotransferase; LDH, lactate dehydrogenase; hs-cTnI, high-sensitivity cardiac troponin I; CRP, C-reactive protein; PFR, PaO_2_/FiO_2_ ratio.

**Figure 3 jcm-13-07106-f003:**
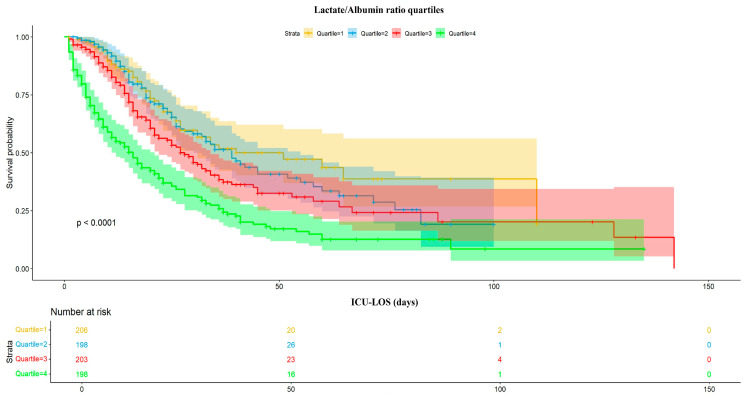
Kaplan–Meier survival curves according to the lactate/albumin ratio quartiles, with shaded ribbons indicating the corresponding 95% CI’s. Abbreviations: ICU-LOS, intensive care unit length of stay; CI, confidence interval.

**Figure 4 jcm-13-07106-f004:**
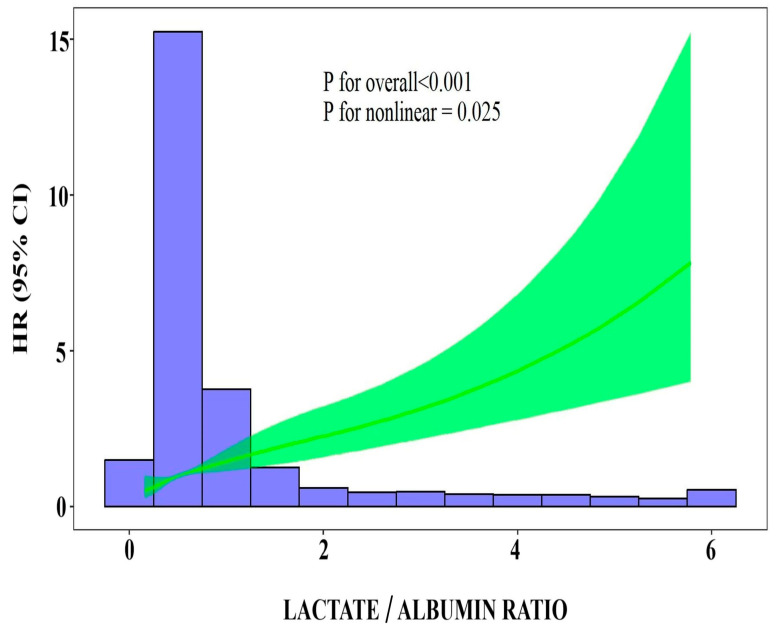
Restricted cubic spline regression analysis of lactate/albumin ratio with ICU mortality. The heavy central lines represent the estimated adjusted HR, with shaded ribbons indicating the corresponding 95% CI. The histogram illustrates the distribution of patients. There was a nonlinear association between lactate/albumin ratio and HR. The model was adjusted for age, sex, NLR, Hb, PLT count, sodium, creatinine, AST, ALT, LDH, hs-cTnI, CRP, fibrinogen, d-dimers, PFR, presence of shock, vaccination status, CRRT, remdesivir, dexamethasone, hypertension, diabetes mellitus, obesity, cardiovascular disease, chronic pulmonary disease, chronic kidney disease, and active malignancy. *Abbreviations*: ICU, intensive care unit; HR, hazard ratio; CI, confidence interval; NLR, neutrophil to lymphocyte ratio; Hb, hemoglobin; PLT, platelets; AST, aspartate aminotransferase; ALT, alanine aminotransferase; hs-cTnI, high-sensitivity cardiac troponin I; CRP, C-reactive protein; PFR, PaO_2_/FiO_2_ ratio; CRRT, continuous renal replacement therapy.

**Figure 5 jcm-13-07106-f005:**
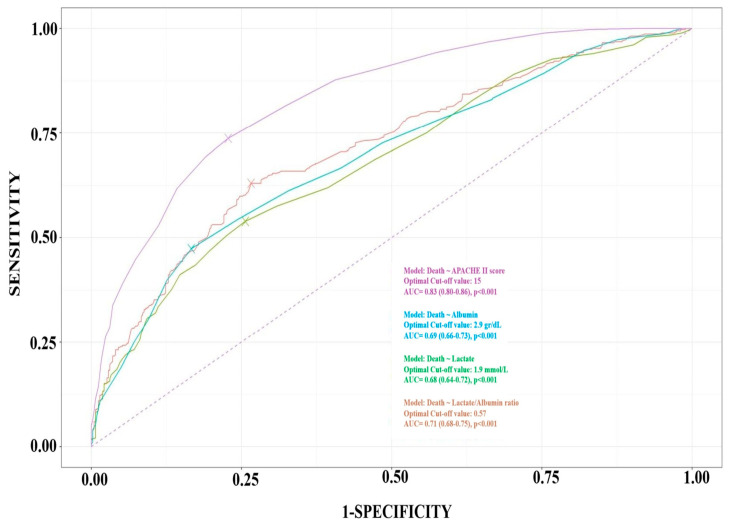
ROC curves for lactate/albumin ratio, lactate, albumin, and APACHE II score in predicting ICU mortality. *Abbreviations*: ROC, receiver– operating characteristics; APACHE, acute physiology and chronic health evaluation; ICU, intensive care unit.

**Figure 6 jcm-13-07106-f006:**
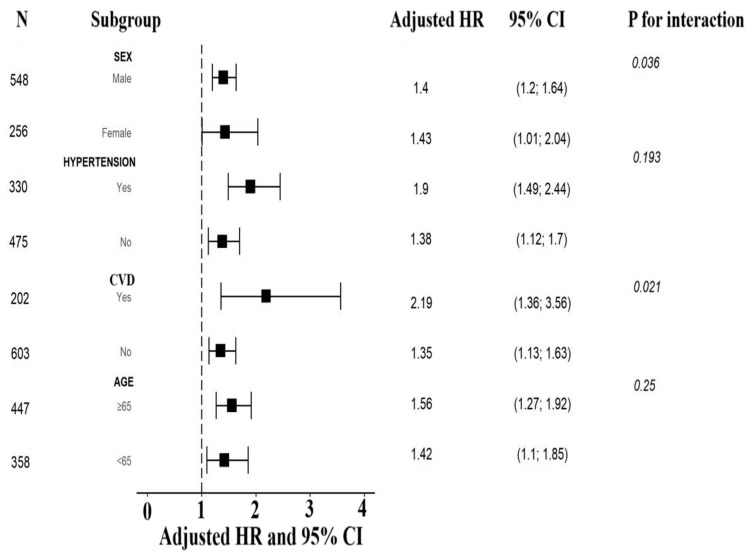
Forest plot of subgroup analyses for the association of lactate/albumin ratio with ICU mortality. Abbreviations: ICU, intensive care unit; HR: hazard ratio; CI: confidence interval; CVD, cardiovascular disease.

**Table 1 jcm-13-07106-t001:** Baseline characteristics of the patient cohort grouped by outcome.

Characteristic	Overall N = 805	Survival N = 421	Non-Survival N = 384	*p*-Value
*Demographics*				
Sex, male, n (%)	548 (68%)	285 (68%)	263 (68%)	0.8
Age, years	67 (57, 76)	61 (52, 70)	73 (65, 79)	<0.001
Age quartiles				<0.001
1	201 (25%)	158 (38%)	43 (11%)	
2	192 (24%)	127 (30%)	65 (17%)	
3	210 (26%)	86 (20%)	124 (32%)	
4	202 (25%)	50 (12%)	152 (40%)	
*Severity scores*				
Charlson comorbidity index	3 (2, 5)	2 (1, 4)	4 (3, 6)	<0.001
APACHE II score	14 (11, 19)	11 (9, 14)	18.0 (14, 23)	<0.001
SOFA score	7 (4, 9)	6 (2, 7)	8 (7, 10)	<0.001
*Outcomes*				
CRRT, n (%)	238 (30%)	51 (12%)	187 (49%)	<0.001
MV duration, days	11 (4, 24)	8 (0, 23)	14 (7, 25)	<0.001
ICU-LOS, days	15 (8, 29)	15 (8, 33)	15 (8, 26)	0.056
*Laboratory tests*				
WBC count, ×10^9^/L	10 (7, 15)	9 (7, 13)	11 (8, 16)	<0.001
Neutrophil count, ×10^9^/L	8.7 (5.7, 12.9)	8.0 (5.4, 11.9)	9.7 (6.6, 14.4)	<0.001
Lymphocyte count, ×10^9^/L	0.76 (0.51, 1.10)	0.80 (0.55, 1.11)	0.73 (0.47, 1.10)	0.015
NLR	12 (7, 20)	10 (6, 17)	14 (8, 23)	<0.001
Hb, g/dL	12.4 (10.8, 13.7)	12.7 (11.4, 13.8)	11.9 (10.1, 13.5)	<0.001
PLT count, ×10^9^/L	241 (182, 309)	254 (199, 321)	226 (164, 296)	<0.001
Albumin, g/dL	3.2 (2.8, 3.6)	3.4 (3.1, 3.6)	3.0 (2.6, 3.4)	<0.001
Sodium, mEq/L	140 (137, 143)	139 (136, 142)	141 (138, 145)	<0.001
Creatinine, mg/dL	0.9 (0.7, 1.3)	0.8 (0.7, 1.0)	1.1 (0.8, 1.6)	<0.001
AST, IU/L	38 (25, 66)	37 (25, 60)	39 (25, 72)	0.2
ALT, IU/L	32 (19, 54)	34 (21, 54)	28 (17, 58)	0.020
LDH, IU/L	456 (347, 624)	437 (312, 573)	488 (370, 669)	<0.001
hs-cTnI, ng/L	24 (10, 77)	15 (7, 41)	42 (17, 123)	<0.001
CRP, mg/dL	11 (5, 18)	10 (4, 17)	12 (7, 20)	<0.001
Fibrinogen, mg/dL	579 (474, 701)	575 (485, 686)	583 (459, 708)	>0.9
D-dimers, mg/L	1.6 (0.9, 3.9)	1.2 (0.7, 2.7)	2.2 (1.1, 5.0)	<0.001
PaO_2_/FiO_2_ ratio	126 (88, 187)	136 (93, 206)	118 (83, 173)	<0.001
Lactate, mmol/L	1.70 (1.30, 2.30)	1.50 (1.20, 1.90)	2.00 (1.45, 2.90)	<0.001
Lactate/albumin ratio	0.53 (0.39, 0.79)	0.46 (0.35, 0.58)	0.64 (0.46, 1.02)	<0.001
Lactate/albumin ratio quartiles				<0.001
1	206 (26%)	150 (36%)	56 (15%)	
2	198 (25%)	123 (29%)	75 (20%)	
3	203 (25%)	96 (23%)	107 (28%)	
4	198 (25%)	52 (12%)	146 (38%)	
*Treatment*				
Full vaccination, n (%)	61 (7.6%)	26 (6.2%)	35 (9.1%)	0.12
Remdesivir, n (%)	391 (49%)	221 (52%)	170 (44%)	0.020
Dexamethasone, n (%)	637 (79%)	328 (78%)	309 (80%)	0.4
Tocilizumab, n (%)	36 (4.5%)	28 (6.7%)	8 (2.1%)	0.002
MV on admission, n (%)	588 (73%)	252 (60%)	336 (88%)	<0.001
HFNC on admission, n (%)	158 (20%)	116 (28%)	42 (11%)	<0.001
Shock on admission, n (%)	304 (38%)	172 (41%)	132 (34%)	0.058
*Comorbidities*				
Hypertension, n (%)	330 (41%)	159 (38%)	171 (45%)	0.051
Diabetes mellitus, n (%)	204 (25%)	99 (24%)	105 (27%)	0.2
Obesity, n (%)	104 (13%)	63 (15%)	41 (11%)	0.070
Cardiovascular disease, n (%)	202 (25%)	81 (19%)	121 (32%)	<0.001
Chronic pulmonary disease, n (%)	106 (13%)	40 (9.5%)	66 (17%)	0.001
Malignancy, n (%)	85 (11%)	24 (5.7%)	61 (16%)	<0.001
Chronic kidney disease, n (%)	65 (8.1%)	15 (3.6%)	50 (13%)	<0.001

Data are expressed as median (IQR), unless otherwise denoted. Abbreviations: IQR, interquartile range; APACHE, acute physiology and chronic health evaluation; SOFA, sequential organ failure assessment; MV, mechanical ventilation; ICU-LOS, intensive care unit length of stay; WBC, white blood cell; NLR, neutrophil to lymphocyte ratio; Hb, hemoglobin; PLT, platelets; AST, aspartate aminotransferase; ALT, alanine aminotransferase; LDH, lactate dehydrogenase; hs-cTnI, high-sensitivity cardiac troponin I; CRP, C-reactive protein; CRRT, continuous renal replacement therapy; HFNC, high-flow nasal cannula.

**Table 2 jcm-13-07106-t002:** Baseline characteristics according to lactate/albumin ratio quartiles.

Characteristic	Quartile 1 N = 206	Quartile 2 N = 198	Quartile 3 N = 203	Quartile 4 N = 198	*p*-Value
*Demographics*					
Sex, male, n (%)	134 (65%)	141 (71%)	139 (68%)	134 (68%)	0.6
Age, years	61 (52, 70)	64 (54, 73)	70 (61, 77)	72 (63, 80)	<0.001
*Severity scores*					
Charlson comorbidity index	3 (1, 4)	3 (1, 4)	4 (2, 5)	4 (3, 6)	<0.001
APACHE II score	11 (8, 16)	13 (9, 16)	14 (12, 18)	19 (15, 24)	<0.001
SOFA score	6 (2, 7)	6 (3, 8)	7 (6, 8)	8 (7, 11)	<0.001
*Outcomes*					
ICU outcome, death, n (%)	56 (27%)	75 (38%)	107 (53%)	146 (74%)	<0.001
CRRT, n (%)	49 (24%)	44 (22%)	65 (32%)	80 (40%)	<0.001
MV duration, days	9 (0, 20)	13 (5, 26)	14 (6, 26)	10 (4, 23)	<0.001
ICU-LOS, days	14 (8, 25)	17 (9, 33)	17 (10, 31)	12 (5, 28)	<0.001
*Laboratory tests*					
WBC count, ×10^9^/L	8 (6, 11)	9 (7, 13)	11 (8, 15)	14 (10, 20)	<0.001
Neutrophil count, ×10^9^/L	6.5 (4.6, 9.5)	8.2 (5.4, 11.9)	9.3 (7.1, 13.7)	12.4 (8.0, 17.4)	<0.001
Lymphocyte count, ×10^9^/L	0.70 (0.49, 1.07)	0.80 (0.54, 1.11)	0.66 (0.48, 0.99)	0.84 (0.54, 1.24)	0.022
NLR	9 (5, 16)	10 (6, 16)	14 (8, 22)	16 (8, 24)	<0.001
Hb, g/dL	12 (11, 14)	13 (11, 14)	12 (11, 14)	12 (9, 13)	<0.001
PLT count, ×10^9^/L	231 (180, 295)	263 (203, 325)	251 (194, 326)	222 (159, 303)	<0.001
Albumin, g/dL	3.6 (3.3, 3.8)	3.3 (3.1, 3.6)	3.0 (2.8, 3.4)	2.8 (2.4, 3.1)	<0.001
Sodium, mEq/L	139 (137, 141)	140 (136, 142)	140 (137, 143)	143 (138, 147)	<0.001
Creatinine, mg/dL	0.8 (0.7, 1.1)	0.8 (0.7, 1.0)	0.9 (0.7, 1.3)	1.2 (0.8, 1.9)	<0.001
AST, IU/L	41 (26, 64)	38 (26, 65)	36 (22, 59)	38 (24, 95)	0.12
ALT, IU/L	34 (21, 54)	33 (21, 55)	30 (19, 51)	29 (15, 67)	0.2
LDH, IU/L	430 (298, 550)	455 (352, 591)	469 (350, 634)	478 (358, 842)	0.001
hs-cTnI, ng/L	13 (7, 38)	16 (8, 47)	23 (12, 62)	74 (27, 223)	<0.001
CRP, mg/dL	11 (4, 18)	11 (5, 17)	11 (5, 18)	13 (6, 21)	0.085
Fibrinogen, mg/dL	595 (513, 726)	588 (506, 679)	593 (485, 721)	510 (375, 669)	<0.001
D-dimers, mg/L	1.1 (0.7, 2.2)	1.3 (0.7, 2.7)	2.2 (1.1, 4.3)	2.5 (1.3, 10.0)	<0.001
PaO_2_/FiO_2_	121 (90, 183)	121 (87, 180)	132 (86, 192)	140 (94, 200)	0.3
Lactate, mmol/L	1.1 (0.9, 1.3)	1.5 (1.4, 1.6)	1.9 (1.7, 2.2)	3.3 (2.7, 5.4)	<0.001
Shock on admission, n (%)	74 (36%)	92 (46%)	94 (46%)	44 (22%)	<0.001
*Treatment*					
Full vaccination, n (%)	15 (7.3%)	12 (6.1%)	14 (6.9%)	20 (10%)	0.5
Remdesivir, n (%)	100 (49%)	109 (55%)	102 (50%)	80 (40%)	0.031
Dexamethasone, n (%)	150 (73%)	166 (84%)	182 (90%)	139 (70%)	<0.001
Tocilizumab, n (%)	5 (2.4%)	11 (5.6%)	13 (6.4%)	7 (3.5%)	0.2
MV on admission, n (%)	104 (50%)	139 (70%)	169 (83%)	176 (89%)	<0.001
HFNC on admission, n (%)	71 (34%)	46 (23%)	28 (14%)	13 (6.6%)	<0.001
*Comorbidities*					
Hypertension, n (%)	84 (41%)	77 (39%)	85 (42%)	84 (42%)	0.9
Diabetes, n (%)	59 (29%)	51 (26%)	47 (23%)	47 (24%)	0.6
Obesity, n (%)	37 (18%)	27 (14%)	29 (14%)	11 (5.6%)	0.002
Cardiovascular disease, n (%)	40 (19%)	42 (21%)	54 (27%)	66 (33%)	0.006
Chronic pulmonary disease, n (%)	25 (12%)	22 (11%)	30 (15%)	29 (15%)	0.6
Malignancy, n (%)	23 (11%)	19 (9.6%)	18 (8.9%)	25 (13%)	0.6
Chronic kidney disease, n (%)	16 (7.8%)	10 (5.1%)	15 (7.4%)	24 (12%)	0.073

Data are expressed as median (IQR), unless otherwise denoted. Abbreviations: IQR, interquartile range; APACHE, acute physiology and chronic health evaluation; SOFA, sequential organ failure assessment; MV, mechanical ventilation; ICU-LOS, intensive care unit length of stay; WBC, white blood cell; NLR, neutrophil to lymphocyte ratio; Hb, hemoglobin; PLT, platelets; AST, aspartate aminotransferase; ALT, alanine aminotransferase; LDH, lactate dehydrogenase; hs-cTnI, high-sensitivity cardiac troponin I; CRP, C-reactive protein; CRRT, continuous renal replacement therapy; HFNC, high-flow nasal cannula.

**Table 3 jcm-13-07106-t003:** Cox regression analysis of lactate/albumin ratio and ICU mortality.

	Univariate Model		* Multivariate Model	
Variable	HR (95%CI)	*p*-Value	HR (95%CI)	*p*-Value
Lactate/albumin ratio	1.60 (1.40–1.70)	<0.001	1.39 (1.27–1.52)	<0.001

* The model was adjusted for age, sex, NLR, Hb, PLT count, sodium, creatinine, AST, ALT, LDH, hs-cTnI, CRP, fibrinogen, d-dimers, PaO_2_/FiO_2_ ratio, presence of shock, vaccination status, CRRT, remdesivir, dexamethasone, hypertension, diabetes mellitus, obesity, coronary, cardiovascular disease, chronic pulmonary disease, chronic kidney disease, and active malignancy. Abbreviations: HR, hazard ratio; CI, confidence interval; NLR, neutrophil to lymphocyte ratio; Hb, hemoglobin; PLT, platelets; AST, aspartate aminotransferase; ALT, alanine aminotransferase; hs-cTnI, high-sensitivity cardiac troponin I; CRP, C-reactive protein; CRRT, continuous renal replacement therapy.

**Table 4 jcm-13-07106-t004:** Predictive characteristics of admission lactate/albumin ratio for ICU outcome.

Lactate/Albumin Ratio Threshold	Sensitivity (%)	Specificity (%)	PPV (%)	NPV (%)
0.43	80	44	56	71
0.57 (optimal cut-off value) *	63	73	68	68
0.64	50	80	70	64

* According to Youden’s index. Abbreviations: ICU, intensive care unit; PPV, positive predictive value; NPV, negative predictive value.

## Data Availability

The datasets used/or analyzed in the present study are available from the corresponding author on reasonable request.

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
