# Peer review of "Lactate to Albumin Ratio and Mortality in Patients with Severe Coronavirus Disease-2019 Admitted to an Intensive Care Unit"

_jcm, 2024, doi:10.3390/jcm13237106_

Round 1
Reviewer 1 Report
Comments and Suggestions for Authors
The study evaluated the effectiveness of the lactate/albumin ratio in predicting ICU mortality in patients with severe COVID-19. A single-center retrospective cohort study included 805 patients, with a median age of 67 and 68% being male. The lactate/albumin ratio was found to be an independent risk factor for ICU mortality, with a hazard ratio of 1.39. The optimal lactate/albumin ratio cut-off for predicting ICU mortality was 0.57, with 63% sensitivity and 73% specificity. The lactate/albumin ratio was significantly associated with mortality across different patient groups. It is essential to clarify specific ambiguities that necessitate elucidation prior to making a decision regarding any publication venue.
1. The lactate/albumin ratio (LAR) serves as an important prognostic marker in sepsis, with research indicating that a higher LAR is associated with elevated mortality rates in critically ill patients. It surpasses lactate levels alone in forecasting inpatient mortality, offering valuable insights into the severity of the condition and possible outcomes. The LAR applies to both adult and pediatric patients, facilitating prompt decision-making in emergency departments.
2. The limitations include that the effect of dynamic changes in the LAR ratio on prognosis has not been evaluated.
3. In my view, the LAR ratio may be appropriate for nearly all patients facing severe disease conditions; however, the authors' selection of it specifically for COVID-19 may not resonate with the reader. They effectively employed appropriate figures and tables to persuade us.
Author Response
Reviewer #1
The study evaluated the effectiveness of the lactate/albumin ratio in predicting ICU mortality in patients with severe COVID-19. A single-center retrospective cohort study included 805 patients, with a median age of 67 and 68% being male. The lactate/albumin ratio was found to be an independent risk factor for ICU mortality, with a hazard ratio of 1.39. The optimal lactate/albumin ratio cut-off for predicting ICU mortality was 0.57, with 63% sensitivity and 73% specificity. The lactate/albumin ratio was significantly associated with mortality across different patient groups. It is essential to clarify specific ambiguities that necessitate elucidation prior to making a decision regarding any publication venue.
- The lactate/albumin ratio (LAR) serves as an important prognostic marker in sepsis, with research indicating that a higher LAR is associated with elevated mortality rates in critically ill patients. It surpasses lactate levels alone in forecasting inpatient mortality, offering valuable insights into the severity of the condition and possible outcomes. The LAR applies to both adult and pediatric patients, facilitating prompt decision-making in emergency departments.
- The limitations include that the effect of dynamic changes in the LAR ratio on prognosis has not been evaluated.
- In my view, the LAR ratio may be appropriate for nearly all patients facing severe disease conditions; however, the authors' selection of it specifically for COVID-19 may not resonate with the reader. They effectively employed appropriate figures and tables to persuade us.
Our Answers
- We thank the Reviewer for positively commenting on the quality of our work.
- This limitation has been already included in the first submitted manuscript (paragraph: limitations)
- We would like to clarify that we focused on the COVID-19 patients exclusively because in this population the lactate/albumin ratio has never been studied.
Reviewer 2 Report
Comments and Suggestions for Authors
The topic from the manuscript is important and the dataset is also substantial, however, the manuscript needs major revision due to following reasons.
1. All figures require higher resolution for better readability
2. Please increase font sizes for axis labels and legends
3. Please use consistent formatting across all figures
4. The statement claiming this is the first study of lactate/albumin ratio in COVID-19 ICU patients is incorrect
5. A thorough literature review should be conducted, including but not limited to:
a. Bergersen et al. (2023) BMC Public Health study on immune and vascular changes (Bergersen, Kristina V., et al. "Health disparities in COVID-19: Immune and vascular changes are linked to disease severity and persist in a high-risk population in Riverside County, California." BMC Public Health 23.1 (2023): 1584.)
b. Other relevant studies on lactate/albumin ratio in COVID-19
6. Figure 3 caption contains two periods, please fix formatting
7. Figure 2
a. Add p-value labels directly on the figure, like put stars in the circles
b. Remove redundant lower triangle of the matrix
8. For the Methods Section
a. Please add raw data preprocessing steps, like how each variable was processed or analyzed
b. Please add statistical software packages and versions used
c. Please add quality control measures
d. Please add data normalization procedures
Author Response
Reviewer #2
The topic from the manuscript is important and the dataset is also substantial, however, the manuscript needs major revision due to following reasons.
We would like to thank you for your comments giving us the chance to clarify the issues you raise and to modify the text in order to improve the quality of this manuscript.
- All figures require higher resolution for better readability
All figures have been revised and resized (resolution=600dpi).
- Please increase font sizes for axis labels and legends
Font sizes of axis labels and legends have been increased in all figures.
- Please use consistent formatting across all figures
Consistent formatting has been used across all figures.
- The statement claiming this is the first study of lactate/albumin ratio in COVID-19 ICU patients is incorrect
Our answer
Indeed, the lactate to albumin ratio has not been studied in patients with COVID-19. To the best of our knowledge, the present study is the first to examine the prognostic performance of this ratio in patients admitted to the ICU due to COVID-19.
- A thorough literature review should be conducted, including but not limited to:
a. Bergersen et al. (2023) BMC Public Health study on immune and vascular changes (Bergersen, Kristina V., et al. "Health disparities in COVID-19: Immune and vascular changes are linked to disease severity and persist in a high-risk population in Riverside County, California." BMC Public Health 23.1 (2023): 1584.)
This article has been included (Reference No. 21) in the revised manuscript.
Please note that all corrections inside the manuscript appear highlighted in red color.
b. Other relevant studies on lactate/albumin ratio in COVID-19
Our answer
Indeed, the lactate to albumin ratio has not been studied in patients with COVID-19.
- Figure 3 caption contains two periods, please fix formatting
We have fixed it.
- Figure 2
a. Add p-value labels directly on the figure, like put stars in the circles
Stars denoting a p-value<0.05 have been added directly on the figure.
b. Remove redundant lower triangle of the matrix
This particular orthogonal format of correlations matrix (although half of it is redundant, as you correctly note) is quite common in literature. Unfortunately, there is no way to modify it, because it has automatically been created by the statistical program R.
- For the Methods Section
a. Please add raw data preprocessing steps, like how each variable was processed or analyzed
As we refer to the first paragraph of the Statistical analysis section: ‘Continuous variables were expressed as median and interquartile range (IQR), while categorical variables were presented as proportions. Continuous variables were analyzed using the Mann-Whitney U test or Kruskal-Wallis test. To handle missing data, we employed multiple imputations, implemented through the 'mice' package in R software. All variables used in the analyses had less than 5% of missing values, except for d-dimers with 19%.’. We also assumed that all variables were non-normally distributed, so we neither checked for normality nor used parametric tests; instead, we used only non-parametric tests to be sure the assumption of normality is not violated.
b. Please add statistical software packages and versions used
We used the latest versions of the following R statistical packages: dplyr, ggplot, mice, gtsummary, survival, survminer, multipleROC, pROC, splines, gridEXTRA. A relevant sentence has been added to the Statistical analysis section.
c. Please add quality control measures
For the Cox regression, the proportional hazards assumption was checked using the Schoenfeld residuals. For the logistic regression, the Hosmer-Lemeshow goodness-of-fit test was used to evaluate whether the logistic regression model was a good fit for the data. A relevant sentence has been added to the Statistical analysis section.
d. Please add data normalization procedures
As we describe in the Statistical analysis section, we used only non-parametric tests for univariate statistics. Moreover, the multivariate tests we used, such as logistic and Cox regression, do not strictly require normal distribution of the variables. Therefore, data normalization procedures were not used.
Round 2
Reviewer 2 Report
Comments and Suggestions for Authors
There are methods to adjust R’s output to display only the non-redundant portion. The following reference provides an example of such a method: https://cran.r-project.org/web/packages/corrplot/vignettes/corrplot-intro.html
The correlation plot appears to have been generated using the R package corrplot, yet this package is not mentioned in the manuscript’s list of tools. Please review all the packages and methods used to ensure completeness and accuracy.
Author Response
Reviewer #2 (Round 2)
There are methods to adjust R’s output to display only the non-redundant portion. The following reference provides an example of such a method: https://cran.r-project.org/web/packages/corrplot/vignettes/corrplot-intro.html.
We would like to thank you very much for your very constructive comments. You have helped us substantially to improve our manuscript. We have used the reference you kindly suggested and created a non-redundant triangular plot (Figure 2). We took the initiative to use an advanced feature of the package, with numbers inside squares representing Spearman’s rho correlation coefficients and blank squares denoting non-significant correlations, e.g. P>0.05. A relevant correction has been made in the figure 2 legend.
The correlation plot appears to have been generated using the R package corrplot, yet this package is not mentioned in the manuscript’s list of tools. Please review all the packages and methods used to ensure completeness and accuracy.
We would like to apologize for this omission. After carefully looking into R history, we found the following:
We used the following R statistical packages: dplyr v. 1.1.4, ggplot2 v. 3.5.1, mice v. 3.16.0, gtsummary v. 2.0.2, corrplot v. 0.94, survival v. 3.6-4, survminer v. 0.4.9, pROC v. 1.18.5, splines v. 4.4.1, ggrcs v. 0.4.1, and gridEXTRA v. 2.3.
A relevant sentence has been added in the Statistical analysis section.
Figure 2 legend
Figure 2. Correlation matrix of lactate/albumin ratio with other variables. Numbers inside squares represent Spearman’s rho correlation coefficients. Blank squares denote non-significant correlations, e.g. P>0.05. Abbreviations: APACHE, acute physiology and chronic health evaluation; SOFA, sequential organ failure assessment; MV, mechanical ventilation; ICU-LOS, intensive care unit-length of stay; WBC, white blood cell; NLR, neutrophil to lymphocyte ratio; Hb, hemoglobin; PLT, platelets; AST, aspartate aminotransferase; ALT, alanine aminotransferase; LDH, lactate dehydrogenase; hs-cTnI, high sensitivity cardiac troponin I; CRP, C-reactive protein; PFR, PaO2/FiO2 ratio.